# WHY DISTILLATION CAN OUTPERFORM ZERO-RL: THE ROLE OF FLEXIBLE REASONING

## ABSTRACT

Reinforcement learning (RL) has played an important role in improving the reasoning ability of large language models (LLMs). Some studies apply RL directly to *smaller* base models (known as zero-RL) and also achieve notable progress. However, in this paper, we show that using only 920 examples, a simple distillation based on the base model can clearly outperform zero-RL, which typically requires much more data and computational cost. By analyzing the token frequency in model outputs, we find that the distilled model shows more flexible reasoning. It uses anthropomorphic tokens and logical connectors much more often than the zero-RL model. Further analysis reveals that distillation enhances the presence of two advanced cognitive behaviors: Multi-Perspective Thinking or Attempting and Metacognitive Awareness. Frequent occurrences of these two advanced cognitive behaviors give rise to flexible reasoning, which is essential for solving complex reasoning problems.

## 1 INTRODUCTION

Recently, large language models have made remarkable progress in reasoning, delivering impressive results in complex mathematical and coding tasks (Jaech et al., 2024; Guo et al., 2025; Team et al., 2025; Qwen Team, 2025; Google DeepMind, 2025). These studies consistently highlight the critical role of reinforcement learning (RL) in their post-training stage. Notably, work such as DeepSeek R1 (Guo et al., 2025) demonstrates that applying RL directly to a *large* base model (DeepSeek-V3-Base 671B (Liu et al., 2024)) without supervised fine-tuning stage (i.e., zero-RL), can lead to substantial performance gains and the emergence of self-reflection reasoning capabilities. Inspired by this promising finding, a growing number of recent studies (Zeng et al., 2025; Li et al., 2025; Yu et al., 2025; Hu et al., 2025; Liu et al., 2025; Yuan et al., 2025) have explored applying zero-RL to *smaller* models (typically those with fewer than 32B parameters). These efforts have also led to noticeable progress on complex mathematical and coding tasks, with emerging patterns of self-reflection observed in the outputs of smaller models.

However, some studies (Guo et al., 2025; Yue et al., 2025) argue that it is more effective to perform distillation rather than zero-RL on small models. In parallel, (Muennighoff et al., 2025; Ye et al., 2025) shows that carefully selecting high-quality prompts and responses for distillation can yield great improvements on complex reasoning tasks as well, even when using only a small amount of data. This naturally raises a series of intriguing questions:

> *Given the same base model (under 32B), can simply fine-tuning it on a small number of high-quality distilled examples match or even outperform zero-RL? And if so, what do these limited distilled examples teach the base model that leads to such improvements?*

In this paper, we focus on the questions outlined above. Firstly, we carefully compare the performance gains of zero-RL and distillation on the same Qwen2.5-32B (Yang et al., 2024a) base model. Specifically, we collect all historical AIME problems (920 in total) from 1983 to 2023 as our prompt set and generate corresponding responses using DeepSeek R1, forming a distilled dataset of 920 examples. We then perform supervised fine-tuning (SFT) on this dataset to obtain the distilled model. Surprisingly, this simple distillation setup is already sufficient to outperform—and in many cases significantly surpass—existing state-of-the-art (SOTA) zero-RL models on reasoning-intensive benchmarks such as AIME, HMMT and GPQA. This result is particularly striking given that zero-RL methods typically rely on orders of magnitude more prompt data and computational resources.

Secondly, we explore why a small amount of distilled data can significantly enhance a model's reasoning ability. By comparing the outputs of the distilled model, zero-RL model, and base model, we find that the distilled model closely mirrors the linguistic style of its teacher, DeepSeek R1, while differing clearly from the zero-RL and base models. The zero-RL model tends to produce formal and rigidly structured responses, often following a fixed step-by-step approach. In contrast, the distilled model shows more flexible reasoning and makes frequent use of anthropomorphic tokens and logical connectors which are rarely seen in zero-RL outputs. These distinctive tokens typically indicate shifts in thinking or reflection on earlier reasoning steps. Notably, when we prevent the distilled model from generating these distinctive tokens during decoding, its performance drops, but remains comparable. This suggests that while these tokens may play an important role in the reasoning process, the distilled model has likely learned more than just surface-level token patterns.

Digging deeper, we find that distillation increases the presence of **two advanced cognitive behaviors**: *Multi-Perspective Thinking or Attempting* and *Metacognitive Awareness*. Frequent occurrences of these two cognitive behaviors give rise to flexible reasoning, which is essential for solving complex reasoning problems, where the solution path is often unclear from the start. The distilled model shows these behaviors far more frequently than the zero-RL model, and a higher frequency of such behaviors is often associated with better reasoning performance. Even when the generation of distinctive tokens (i.e., those anthropomorphic tokens and logical connectors mentioned before which differ between the distilled model from the zero-RL model outputs) is blocked during decoding, the distill model actively tries to work around the restriction to express these behaviors. This indicates that the distilled model has internalized these cognitive behaviors at a deeper level from its teacher model.

Finally, we discuss potential issues of reward hacking and overfitting observed in zero-RL model outputs, the limitation of zero-RL for complex reasoning, as well as the possibility of achieving better performance through subsequent RL scaling.

## 2 RELATED WORK

**Zero-RL.** Reinforcement Learning (RL) has been shown to improve LLM's reasoning capability (Qwen Team, 2025; Guo et al., 2025; Lai et al., 2024). Conventionally, RL requires the initial policy to be firstly fine-tuned with task-related data, since the output of the base model on a specific task can be disorganized or incoherent (Ouyang et al., 2022; Bai et al., 2022). However, recent works demonstrate that, starting from *the base model*, RL algorithms (e.g., PPO (Schulman et al., 2017), GRPO (Shao et al., 2024)) using rule-based reward can greatly improve its reasoning ability and even trigger the "Aha moment" (Guo et al., 2025; Yu et al., 2025; Hu et al., 2025). Such methods that directly conduct RL with base model are referred to as zero-RL, meaning starting from "zero" (the base model). These zero-RL models are typically trained with tens of thousands of prompt samples and optimized for thousands of steps. The prompt samples typically need to be carefully selected to match the capabilities of base model and provide sufficient challenge; otherwise, performance gains from zero-RL tend to saturate (Zeng et al., 2025; Hu et al., 2025; Yuan et al., 2025; Yang et al., 2025).

**Distillation from reasoning model.** Several methods tried to elicit LLM's reasoning capability through model distillation (Ye et al., 2025; Muennighoff et al., 2025; Labs, 2025; Xu et al., 2025). Specifically, these methods first pre-collect full responses on complex reasoning problems from strong existing reasoning models (e.g., DeepSeek-R1, QwQ-32B). Subsequently, they conduct supervised fine-tuning (SFT) with these responses. Previous methods in this line often use carefully curated questions and responses (Ye et al., 2025; Muennighoff et al., 2025). By contrast, we construct the dataset for distillation with a single data source and a single reasoning model, and without any data filtration. We then perform supervised fine-tuning with *the base model*, also using these limited data.

## 3 CASE STUDY: DISTILLATION USING 920 SAMPLES

We choose Qwen2.5-32B (Yang et al., 2024a) as the base model and compare two approaches built on top of it: zero-RL using a larger prompt dataset, and distillation using a small set of outputs from a reasoning teacher model: DeepSeek R1 (Guo et al., 2025). We also present distillation experiments using other teacher models in Appendix C.2, and the results lead to similar conclusions.

**Zero-RL models**. Since prior work has already conducted extensive experiments, we directly consider three open-source models that currently achieve state-of-the-art (SOTA) performance on zero-RL with Qwen2.5-32B: DAPO-32B (Yu et al., 2025), Open-Reasoner-Zero-32B (i.e. ORZ-32B) (Hu et al.,

2025) and SimpleRL-Zoo-32B (i.e. SimpleRL-32B) (Zeng et al., 2025)[1]. These models are typically trained on tens of thousands of carefully selected prompt samples and optimized over thousands of training steps. For each prompt, the algorithm often needs to generate more than 16 responses to ensure a mix of correct and incorrect answers for effective gradient updates. This process generally requires much more forward and backward passes than standard SFT.

**Distilled models**. Recent work such as s1 (Muennighoff et al., 2025) and LIMO (Ye et al., 2025) also emphasizes that using a small amount of carefully selected, high-quality distillation data can lead to significant performance improvements. However, these studies are based on *Qwen2.5-Instruct*, which has typically already undergone RL. To enable a fair comparison with zero-RL models, we conduct distillation experiments on the Qwen2.5-32B base model using the historical AIME problems, without *any deliberate filtering or selection*, in order to maintain a simple distillation setup.

Specifically, we construct the dataset by collecting all 920 AIME problems from 1983 to 2023 and generating one reasoning response for each using DeepSeek R1. This yields a distillation dataset in which each problem is paired with a DeepSeek R1-generated solution. DeepSeek R1 achieves an overall accuracy of 85.4% on this dataset. We do *not* filter for correctness; instead, we retain all samples regardless of whether the answers are correct or not. We then perform SFT on Qwen2.5-32B using this dataset for 5 epochs to obtain the distilled model. We use the prompt template from Qwen2.5-Math (Yang et al., 2024b) for training. For more details about the training setup and computational resource usage, see Appendix B.1, B.2, B.3.

**Evaluation settings**. We evaluate the performance of the two approaches on five challenging benchmarks: AIME2024 (part I, 2024; part II, 2024), AIME2025 (part I, 2025; part II, 2025), HMMT Feb 2025 (MathArena Team, 2025), GQPA Diamond (Rein et al., 2024), and MATH500 (Hendrycks et al., 2021). AIME 2024 and AIME 2025 represent the American Invitational Mathematics Examination held in 2024 and 2025. AIME is an Olympiad-style exam that tests a wide range of mathematical abilities. It contains 30 problems each year. HMMT is one of the largest and most prestigious high school competitions. HMMT February 2025 contains 30 challenging problems. GPQA is a benchmark designed to evaluate the capabilities of LLMs in tackling challenging scientific questions. It consists of 448 multiple-choice questions carefully crafted by experts in biology, physics, and chemistry. GPQA Diamond is its most challenging subset that contains 198 questions. MATH500 is a benchmark of math problems selected by OpenAI (Hendrycks et al., 2021).

To ensure accurate and fair evaluation, we carefully consider parameters that could influence the results to guarantee reproducibility (Hochlehnert et al., 2025). We set the evaluation temperature to 1, top-p to 0.95, and the maximum generation length to 32,768. For open-source zero-RL models, we use the prompt templates specified in their original papers. For our distilled models, we use the same prompt template as in training. For the Qwen2.5 base model, we use no prompt template, as we find this setting clearly outperforms alternative prompts. All models are evaluated using the official evaluation code from Qwen2.5-Math (Yang et al., 2024b) to ensure consistency and fairness. Considering the potential impact of prompt templates (Yang et al., 2025) and sampling parameters such as temperature, we report additional results under alternative settings in the Appendix C.2. For AIME and HMMT, we report Avg@32 (i.e., the average Pass@1 results over 32 independent runs), as well as Pass@8$^{(40)}$ [2]. For GQPA Diamond and MATH500, we report Avg@8 (i.e., the average Pass@1 results over 8 independent runs).

**Evaluation results**. As shown in Table 1, the distilled model which trained on only 920 examples, consistently outperforms zero-RL models. The distilled model also achieves notably better performance than the base model. Additionally, the distilled model produces significantly longer responses.

This result is surprising to us. In general, SFT with only 920 samples would not be expected to introduce new knowledge; nevertheless, it produces a substantial improvement in performance. Although the number of training samples between distillation and zero-RL is *not directly comparable*, since distillation data includes teacher model outputs, the effectiveness of this simple distillation setup is still striking. The training samples used in some zero-RL methods (Hu et al., 2025) typically

---

[1]For all compared open-source zero-RL models, we use the latest publicly released versions.

[2]To achieve more unbiased estimation, we report Pass@8 using unbiased estimator in (Chen et al., 2021) (i.e., Pass@$k = 1 - \binom{n-c}{k}/\binom{n}{k}$), computed over 40 model responses per problem per model. We denote this as Pass@8$^{(40)}$.

Table 1: Performance of different models across benchmarks. Avg@32 denotes the average Pass@1 score over 32 independent runs. AIME and HMMT is evaluated using both Avg@32 and Pass@8, while GPQA Diamond and MATH500 are evaluated using Avg@8.

| Metric | Distilled -32B | Zero-RL (DAPO-32B) | Zero-RL (ORZ-32B) | Zero-RL (SimpleRL-32B) | Qwen2.5 -32B-base |
|---|---|---|---|---|---|
| # of training samples | **920** | 17,000 | 57,000 | 8,000 | - |
| AIME2024 (Avg@32) | **61.2** | 50.6 | 41.9 | 27.3 | 16.8 |
| AIME2024 (Pass@8[(40)]) | **82.7** | 71.3 | 65.9 | 48.7 | 46.9 |
| AIME2025 (Avg@32) | **50.0** | 32.9 | 33.3 | 10.2 | 8.3 |
| AIME2025 (Pass@8[(40)]) | **74.7** | 51.7 | 53.4 | 28.1 | 27.9 |
| HMMT Feb 2025 (Avg@32) | **34.6** | 13.8 | 20.9 | 5.4 | 1.9 |
| HMMT Feb 2025 (Pass@8[(40)]) | **65.0** | 28.3 | 38.3 | 9.3 | 10.0 |
| GPQA Diamond (Avg@8) | **60.0** | 48.7 | 57.7 | 48.4 | 34.9 |
| MATH500 (Avg@8) | **93.8** | 68.0[*] | 90.7 | 89.2 | 70.1 |
| Avg. Length (AIME2024) | 13975 | 7916 | 10174 | 1182 | 1148 |
| Avg. Length (AIME2025) | 15034 | 6610 | 9522 | 1298 | 1088 |
| Avg. Length (HMMT Feb 2025) | 16609 | 11978 | 10940 | 1190 | 969 |
| Avg. Length (GQPA) | 10237 | 5073 | 7808 | 823 | 565 |
| Avg. Length (MATH500) | 4239 | 5250 | 4230 | 662 | 603 |

[*] Relatively low score on this benchmark may be due to DAPO's requirement for integer-only answers during RL training. See the Discussion section 5 and Appendix C.2 for more details.

include a subset of historical AIME problems (i.e., our distillation prompts) to provide challenging tasks. Considering the large gap in the number of samples, training steps, and computational cost (see Appendix B.3 for more detailed comparisons), the gains achieved through distillation are unexpectedly impressive. Beyond the challenging mathematical benchmarks discussed above, we also find that the distilled 32B model performs strongly in benchmarks for other domains (see Appendix C.2), whereas some zero-RL models appear more prone to reward hacking and overfitting.

To better understand the underlying factors, we aim to answer the following questions: *What did the 920 distilled examples from DeepSeek R1 teach the base model? Why do zero-RL models, despite being trained on tens of times more data, still fail to outperform the distilled model?* In the following section, we firstly examine the linguistic patterns in model outputs to uncover the distinct reasoning pathways produced by the distilled and zero-RL models.

## 4 Two Distinct Reasoning Pathways

### 4.1 Linguistic Patterns of Distilled vs. Zero-RL Model Outputs

Taking the problems from AIME2024 as the examples, we firstly compare the outputs of the two types of models in terms of linguistic patterns. For clarity, in this section we focus on the distilled model introduced in Section 3 and the DAPO-32B zero-RL model. Similar patterns are observed in other zero-RL models, with full statistics reported in the Appendix D.1.

From an overall perspective, we can observe two distinct styles in the models' solutions, as illustrated in Table 2. The zero-RL model tends to produce more formal and structured responses, typically following a "step-by-step" approach to solving the problems. The model appears to be fairly confident about the solution path from the beginning and proceeds step by step along a predetermined framework, with occasional verification of earlier steps. In contrast, the distilled model's outputs resemble the thought process of a human solver, *frequently* involving trial and error, revisiting earlier steps, and expressing uncertainty throughout the reasoning process. These behaviors are often reflected through expressions of hesitation or uncertainty, such as "wait," "perhaps," and "maybe"; phrases indicating a shift in strategy, like "try another idea" and "alternatively"; and acknowledgments of mistakes, such as "it's wrong here." Through this iterative exploration, the model gradually converges on the correct answer. This phenomenon is similar to the difference observed between "aha" and non-"aha" model outputs in recent work (Yang et al., 2025).

To systematically analyze the differences between the two models' outputs, we perform a token frequency analysis. We modify the token categorization approach introduced by (Yang et al., 2025), defining three token types: anthropomorphic tokens, logical connectors, and mathematical reasoning

Table 2: The contrasting solution styles of the two models on an example from AIME 2024.

| |
| --- |
| **Question:** Define $f(x) = \left| \, |x| - \frac{1}{2} \, \right|$ and $g(x) = \left| \, |x| - \frac{1}{4} \, \right|$. Find the number of intersections of the graphs of $y = 4g(f(\sin(2\pi x)))$ and $x = 4g(f(\cos(3\pi y)))$. |

| **Zero-RL model (DAPO-32B) response:** | **Distilled model response:** |
| --- | --- |
| To approach this math problem step by step... | Okay, so I need to find... **Hmm**, that sounds a bit complicated ... |
| ... | ... |
| Let's break down the problem **step-by-step**. | **Wait**, if $x$ is between $-\frac{1}{2}$ and $\frac{1}{2}$, then $f(x) = \frac{1}{2} - |x|$. If $|x| \geq \frac{1}{2}$, |
| **Step 1:** Analyze the function $f(x) = \left| \, |x| - \frac{1}{2} \, \right|$. | then ... |
| **Step 2:** Analyze the function $g(x) = \left| \, |x| - \frac{1}{4} \, \right|$. | ... |
| Now, we need to compute $g(f(u))$ for some input $u$. First, let's find | **Wait, perhaps another way:** For each period of $\sin(2\pi x)$ ... |
| $f(u)$ and then plug it into $g$: | ... |
| 1. Compute $f(u) = \left| \, |u| - \frac{1}{2} \, \right|$. | **Wait, hold on** ... **maybe my approach is wrong** here. **Wait, perhaps** |
| Then we plug this result into $g$: | **an easier way**... |
| | ... |
| $$g(f(u)) = g\left( \left| \, |u| - \frac{1}{2} \, \right| \right) = \left| \, \left| \, |u| - \frac{1}{2} \, \right| - \frac{1}{4} \, \right|$$ | **Alternatively**, since both functions are composed of periodic... |
| | ... |
| ... | **But** I need a better strategy ... **here's an idea**... but I'm not confident... |
| [Answer]. | ... |
| | [Answer]. |

tokens: (1) **Anthropomorphic tokens** include words like "okay" "me" "wait" and "hmm" as well as uncertain terms like "perhaps" and "maybe" and conversational phrases such as "hold on". In the context of the problem-solving process, these tokens typically indicate hesitation or uncertainty during reasoning. (2) **Logical connectors** refer to words such as "but" "however" and "alternatively" which signal contrast, progression, or coordination in problem-solving process. (3) **Mathematical reasoning tokens** include terms like "define" "denote" "imply" and "simplify" which commonly appear in written mathematical solutions. The detailed token categorization and the rationale behind it are provided in Appendix D.2.

We specifically analyze the token frequencies of the three categories across the full responses of each model. As shown in Figure 1, the distilled model uses anthropomorphic language and logical connectors much more often than the zero-RL model. All the anthropomorphic words like "wait" and "maybe" appear often in the distilled model's responses but are almost never seen in those from the zero-RL model. The distilled model also makes greater use of logical connectors—especially words like "but," "therefore," and "alternatively." The word "alternatively," which often signals a shift in approach or line of thinking, is nearly absent from the zero-RL outputs. This may suggest that the distilled model tends to explore alternative ideas more actively and shift its reasoning direction more frequently. Figure 1 also shows that both models use a similar amount of mathematical reasoning tokens, while the total count is slightly higher in the outputs of the zero-RL model.

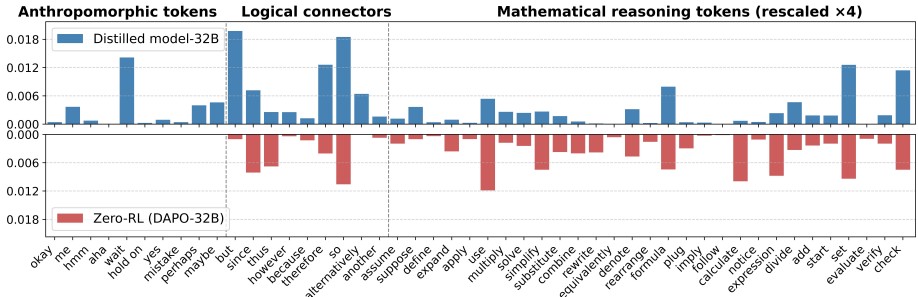

Figure 1: Comparison of token usage between the Distilled and zero-RL models responses to AIME2024 problems across anthropomorphic tokens, logical connectors, and mathematical reasoning tokens. The mathematical reasoning tokens are rescaled by a factor of 4 for better visibility.

We also performed a token frequency analysis on the base model, Qwen2.5-32B-base, using its responses to the AIME2024 problems. As shown in Figure 2, the base model shows a response pattern very similar to that of the zero-RL models built on top of it—mainly following a step-by-step approach, with very few anthropomorphic tokens and limited use of logical connectors. Zero-RL models show some differences from the base model in their use of certain mathematical reasoning tokens,

suggesting that RL may adjust the probabilities of these tokens based on the base model's behavior. However, for tokens that rarely appear in the base model, such as anthropomorphic expressions or those that reflect shifts in reasoning (e.g., alternatively), RL doesn't seem to significantly increase their usage.

Figure 3 shows the token frequency in the responses of the teacher model, DeepSeek R1. The distribution shows a clear resemblance to that of the distilled model, particularly in the use of anthropomorphic tokens and logical connectors. This suggests that, at the token level, the distilled model may have learned to imitate its teacher, DeepSeek R1, whose reasoning style is likely more effective and expert-like.

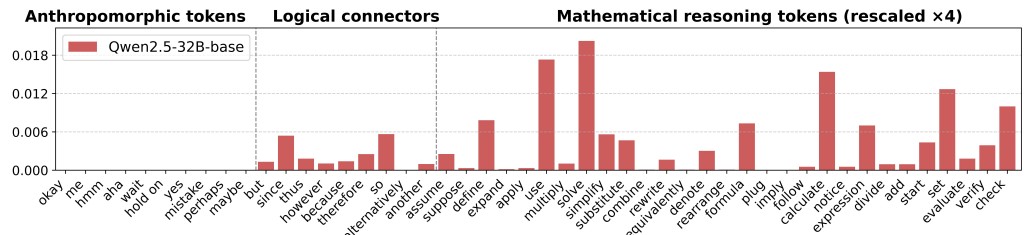

Figure 2: Token usage in Qwen2.5-32B-base's responses to AIME2024 problems across anthropomorphic tokens, logical connectors, and mathematical reasoning tokens.

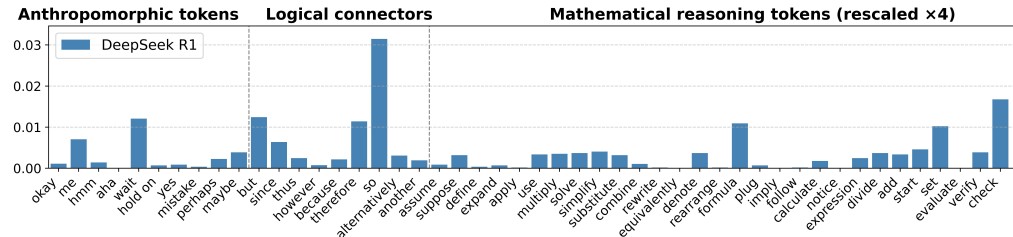

Figure 3: Token usage in DeepSeek R1's responses to AIME2024 problems across anthropomorphic tokens, logical connectors, and mathematical reasoning tokens.

**What if the distilled model is prevented from generating these distinctive tokens?**

Since these anthropomorphic tokens and logical connectors are linguistic features learned by the distilled model from the teacher model and are largely absent in the zero-RL and base models, we would like to know what happens to the distilled model's performance if it is prevented from generating these *distinctive tokens*[3] during decoding.

We select the tokens with the largest frequency differences between the distilled model and the zero-RL model as shown in Figure 1, including words such as "wait," "me," "perhaps," "maybe," "alternatively," and "but," and prevent the distilled model from generating them during decoding. The full list of banned tokens is provided in Appendix D.3. Table 3 shows a clear performance drop for the distilled model when these distinctive tokens are banned, across all benchmarks. This suggests that anthropomorphic tokens and logical connectors play an important role in enhancing the model's reasoning performance. For difficult problems, the performance drop is larger. For example, on AIME2025, the score drops by 28.2%, suggesting that harder problems may rely more heavily on the reasoning patterns enabled by these tokens.

It is worth noting that although the performance of the token-restricted distilled model decreases, it still notably outperforms the base model and remains comparable to zero-RL model. Interestingly, we observe that the model actively tries to work around the banned token constraints, using other expressions or other tokens to convey shifts in reasoning and awareness of potential errors in the

---

[3]For simplicity, *distinctive tokens* refer to the anthropomorphic tokens and logical connectors mentioned in subsection 4.1 which differ between the distilled model and the zero-RL model outputs

Table 3: Performance drop of the distilled model when prevented from generating distinctive tokens. Distilled-32B (Token-Restricted) refers to the distilled model with generation of these tokens *disabled* during decoding.

| Metric | Distilled-32B | Distilled-32B (Token-Restricted) | $\Delta$ |
|---|---|---|---|
| AIME2024 (Avg@32) | **61.2** | 50.3 | -10.9 |
| AIME2025 (Avg@32) | **52.9** | 38.0 | -14.9 |
| HMMT Feb 2025 (Avg@32) | **34.6** | 26.4 | -8.2 |
| GPQA Diamond (Avg@8) | **60.0** | 56.0 | -4.0 |
| MATH500 (Avg@8) | **93.8** | 91.7 | -2.1 |

solution process. This implies that the distilled model may have learned more than just **surface-level token patterns**—it has picked up some **deeper reasoning behaviors** from the teacher model!

In the next subsection, we take a closer look at the advanced cognitive behaviors introduced by distillation. These behaviors reflect how humans tackle complex and unfamiliar problems, and are likely important for solving difficult reasoning tasks.

## 4.2 ANALYZING ADVANCED COGNITIVE BEHAVIORS

Existing study (Gandhi et al., 2025) mentions that four types of cognitive behaviors, namely back-tracking, verification, subgoal setting, and backward chaining, are highly beneficial for solving reasoning problems. This is also considered one of the reasons why the Qwen series of models often achieve strong performance. Both the Qwen2.5-32B base model and its zero-RL variants in our experiments can exhibit these cognitive behaviors.

However, when it comes to solving challenging reasoning problems or tasks that require creative thinking, such as competition problems in AIME, it is often difficult to fully plan out a solution path from the beginning. Rigidly following a "step-by-step" approach, can easily lead to overconfidence in suboptimal directions. Techniques like subgoal setting and backward chaining are valuable, but what matters more is using them flexibly within a process of exploring and testing multiple ideas.

Let us consider how humans approach difficult or unfamiliar problems. A skilled solver may begin by applying familiar strategies, but when stuck, they quickly shift perspective and explore alternative angles, continuously trying new ideas. Throughout the process, mistakes are common, and there is often considerable uncertainty about whether the current approach is on the right track. Hence, they frequently check for errors and reflect on their reasoning. In general, the path to the correct solution involves a repeated cycle: trying an idea, checking for mistakes, identifying errors, learning from them, and then attempting the next idea. Building on this intuition, we introduce two *advanced* cognitive behaviors that we believe are especially important for solving such difficult reasoning tasks:

- **Multi-Perspective Thinking or Attempting**: Viewing a problem from diverse perspectives to gain fresh insights, or exploring different ideas and alternative approaches to make meaningful progress.

- **Metacognitive Awareness** (Schraw & Dennison, 1994): Actively reflecting on your reasoning process during problem-solving to assess progress, evaluate current strategies, and identify potential errors in real time. Behaviors such as reflective hesitation, backtracking, and verification are all integral components of this awareness.

Frequent occurrences of these two advanced cognitive behaviors give rise to flexible reasoning. Both of the advanced cognitive behaviors are reflected through certain key phrases, which can be interpreted in context. For example, expressions like "let's try another angle..." or "but I need a better strategy ... here's an idea, let's try... <solving process>..." often indicate *Multi-Perspective Thinking or Attempting*; and expressions such as "wait, maybe my approach is wrong here" or "it seems not correct, step back" typically indicate *Metacognitive Awareness*. We use GPT-4o (Hurst et al., 2024) to identify the number of occurrences of advanced cognitive behaviors in model responses. Specifically, for each model's response to each problem, we prompt GPT-4o to identify which parts of the response reflect either of the two advanced cognitive behaviors, and count how many times each behavior appears per response (as they often occur more than once). The detailed prompt template and additional statistics are provided in Appendix D.4.

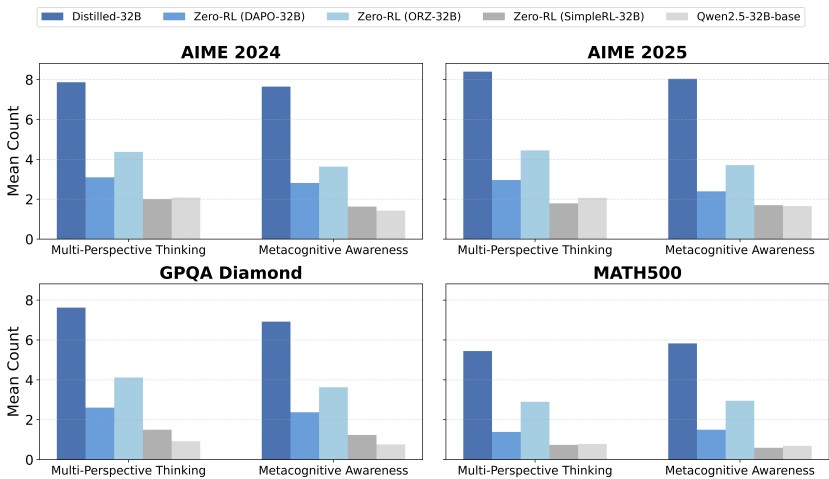

Figure 4: Comparison of the number of advanced cognitive behaviors per response across benchmarks. Additional result are provided in Appendix D.4.

Figure 4 shows the average number of advanced cognitive behaviors exhibited by the distilled model, zero-RL models, and the base model across the four benchmarks. The distilled model clearly demonstrates more frequent use of both behaviors compared to the others. Across models, we observe a **strong correlation** between **the number of cognitive behaviors** and **benchmark performance** (Table 1): the distilled model shows the highest behavior counts and benchmark scores, while the base model and SimpleRL-32B show both lower behavior counts and lower benchmark scores. Compared to distilled model, the zero-RL *fails* to significantly boost the frequency of the two behaviors over the base model, even though some zero-RL models have already trained extensively with large computational resources over multiple epochs or thousands of steps.

As discussed in Section 4.1, certain tokens (i.e. anthropomorphic tokens and some logical connectors) show clear frequency differences between the distilled model and the zero-RL model. When preventing from generating these tokens, the performance of distilled model drops but still remains comparable to that of the zero-RL model. The underlying reason is that, even when restricted from using these distinctive tokens, the distilled model still actively seeks to express the two advanced cognitive behaviors **through other means**. We present a quite interesting example in Table 4. Wwhen the distilled model is prevented from generating banned tokens such as "wait" or "alternatively," it works around the restriction by switching to *Chinese* to express similar thinking shifts and error recognition, which still reflects the two advanced cognitive behaviors.

## 5 DISCUSSION

**Potential reward hacking and overfitting in zero-RL.** Works such as (Yu et al., 2025; Hu et al., 2025; Zeng et al., 2025) have contributed valuable open-source datasets and provided detailed training reports. However, we still identify several cases that may indicate reward hacking or overfitting. One such example is shown in Table 5. In this MATH500 problem, ORZ-32B generates a relatively long reasoning trace but abruptly ends the reasoning process right after the token "`let's`," outputting the end-of-thought marker `</think>` and "guessing" a (wrong) answer, even though the response had not yet reached the maximum generation length we set. We check the length of the output and found it to be close to the training-time maximum, suggesting that this behavior might have accidentally led to a correct answer during training and is reinforced through reward. DAPO-32B first generates a fractional result, but when producing the final answer, it outputs an integer. We notice that all answers in training set are integers, which suggests that the model may overfit to the expectation that only integer answers are correct, due to consistent negative reward for non-integer outputs. These phenomena can be frequently observed in the outputs of zero-RL models, suggesting we still need to be careful when choosing RL parameters, including details such as the maximum generation length during training and the format of ground truth answers in the data.

**The limitation of recent zero-RL for complex reasoning.** As we present in subsection 4.2, even though some zero-RL models have already trained extensively with large computational resources

Table 4: The comparison of distilled model and token-restricted distilled model on an example from AIME 2024. Token-restricted distilled model refers to the distilled model with generation of these distinctive tokens *disabled* during decoding.

| **Question:** Let $N$ be the greatest four-digit positive integer with the property that whenever one of its digits is changed to 1, the resulting number is divisible by 7. Let $Q$ and $R$ be the quotient and remainder, respectively, when $N$ is divided by 1000. Find $Q + R$. | |
|---|---|
| **Distilled model response:**
Okay, so I need to find...
...
$(6A + 2B + 10 + D) \equiv 0 \bmod 7$ (Wait, hold on. 10 mod 7 is 3, so $10 \equiv 3 \bmod 7$. Wait, but 10 is 3 mod 7, **right? Let me verify**...
...
**Wait**, but we need to **check** if all four original congruences hold.
...
Check equation (d): $6A + 2B + 3C = 6 * 5 + 2 * 6 + 3 * 2 = 30 + 12 + 6 = 48$. 48 mod 7=6. Equation (d) requires 6. Correct. **Wow!** So all equations are satisfied with A=5, B=6, C=2, D=4. Therefore, N=5624.
**But wait**, this is for $A = 5$. But $N = 5624$. However, **maybe** higher A values also work? **Wait**, we checked $A = 9, 8, 7, 6, 5$. For $A = 6$, equation (a) didn't hold; $A = 5$ works.
...
**Alternatively**, if there are other solutions where digits are higher despite different residues?
...
Thus, N=5694. There's no higher digits possible. Hence, this is the maximum.
...
[Answer]. | **Token-restricted distilled model response:**
Alright, so I need to find...
...
First, let's parse the problem. N is a four-digit number...
...
Our goal is to find the largest such $N$. Since we want the largest four-digit number, we should start checking from 9999 downwards, 但这个方法可能很耗时。可能有更好的方法。
...
Now, **check** if these numbers satisfy all four original equations...
...
Since we need the greatest such number, between 5694 and 5624, it's 5694. 但是，这里有个问题，题目说N是最大的四位数。但是5694和5624都是5开头，可能还有更大的数。例如，6开头或者7开头的数。但我们刚才得到a=5。怎么回事？
...
但等一下，我们可能漏掉了某些情况。因为当计算d的时候...
...
但是，我们需要确保没有更大的N，比如说6000多的数。为什么我们的解得出a=5？这是因为我们在解同余方程时...
...
[Answer]. |

Table 5: An example of zero-RL outputs to a MATH500 problem.

| **Question:** Let $p(x)$ be a polynomial of degree 5 such that $p(n) = \frac{n}{n^2-1}$ for $n = 2, 3, 4, \ldots, 7$. Find $p(8)$. | |
|---|---|
| **Zero-RL model (ORZ-32B) response:**
`<think>`
...
There is a sign error, so let's double-check the setup. Revisiting the expressions, **let's**
`</think> <answer>`
$$\boxed{\frac{8}{63}}$$
`</answer>` | **Zero-RL model (DAPO-32B) response:**
...
Finally, after the detailed process above, we found $p(8)$ by the Lagrange interpolation method to be $\frac{3}{56}$.
Answer: $\boxed{\frac{3}{56}}$
Let's convert this to the final form of the answer.
It seems the solution to the problem yields the polynomial value of
$p(8) = \boxed{\frac{3}{56}}$.
Answer: **3** |

over multiple epochs or thousands of steps, they still cannot significantly boost the frequency of advanced cognitive behaviors. For complex reasoning task, the two cognitive behaviors are key and effective reasoning patterns. While the distilled model can acquire them directly through distillation, zero-RL struggles to *identify* and reinforce these patterns even when the final policy entropy has already decreased a lot. Similarly, for other specific downstream tasks, appropriate distillation or SFT to bring the important patterns to base model may still be cost-effective and necessary.

**Toward better performance via subsequent RL scaling.** In this paper, we emphasize the value of distillation from a teacher model. This does not mean RL is ineffective; rather, we believe that distilled models are better suited for subsequent RL. Distillation introduces advanced cognitive behaviors that enable more diverse reasoning paths, which may help RL extract richer feedback signals. We believe it is a promising pathway toward reproducing models like OpenAI o1 or DeepSeek R1. For further discussion, see Appendix E.

## 6 CONCLUSION

In this paper, we find that the distilled model, which shows impressive reasoning ability, generates much more anthropomorphic tokens and logical connectors compared to the zero-RL model. Going further, we observe that distillation enhances two advanced cognitive behaviors in the base model: Multi-Perspective Thinking or Attempting, and Metacognitive Awareness, which appear to be key factors in improving reasoning ability.

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

## A  LIMITATION

Our work also has some limitations. First, our work highlight the importance of distillation for relatively *smaller* model. In this paper, we use 32B model as our base model. Future work should extend the investigation to medium-sized models, such as 70B, to further explore the manifestation and impact of the two advanced cognitive behaviors. Likewise, smaller models below 32B should also be studied in depth. This is already part of our planned future research.

Second, although we have demonstrated that the distilled model exhibits the two advanced cognitive behaviors introduced in subsection 4.2, there may be other advanced reasoning behaviors learned from the teacher model that are not covered in this paper. For example, we observe that the distilled model tends to abstract the problem and connect it with prior knowledge to find potential breakthroughs, an ability that is also important for solving complex reasoning problems. However, since this behavior appears less frequently than the two cognitive behaviors introduced in subsection 4.2, and because we find that using GPT-4o to identify this behavior is highly unstable, we do not explicitly include it in this work. Future work should investigate these additional advanced cognitive behaviors and use more advanced models and methods for reliable identification.

## B  EXPERIMENTAL DETAILS

### B.1  DETAILS OF DISTILLATION DATA

To construct the distillation dataset, we use the reasoning model DeepSeek R1 (Guo et al., 2025) to generate responses for all 920 AIME problems from 1983 to 2023. DeepSeek R1 achieves an overall accuracy of 85.4% on this set. We directly use the problem-response pairs without any filtering based on correctness or prompt content. The distribution of DeepSeek R1 response length is shown in Figure 5.

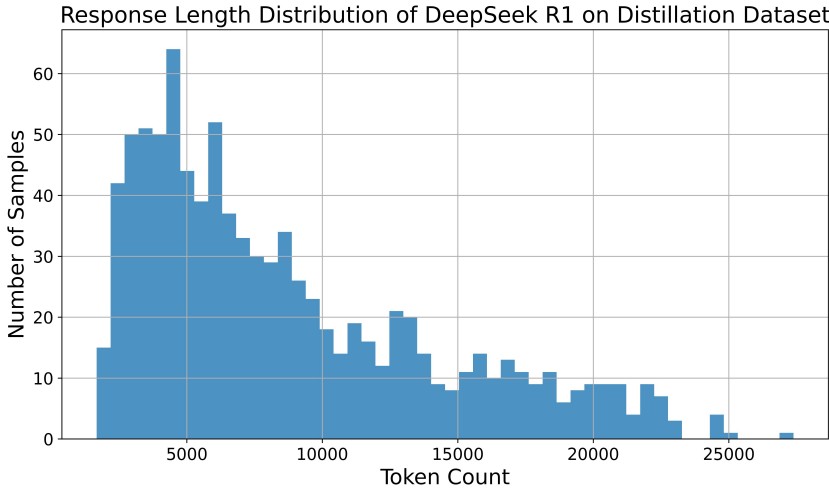

Figure 5: Response length distribution of DeepSeek R1 on 920 distillation problems.

### B.2  TRAINING DETAILS OF DISTILLATION

We use the prompt template from (Yang et al., 2024b) for distillation. See Table 6 for details. We train using bfloat16 precision. The learning rate is linearly increased to 1e-5 over the first 5% of training steps, then decayed to zero following a cosine schedule for the rest of training. See Table 7 for detailed training configurations. The training framework is based on the implementation in (Muennighoff et al., 2025).

The learning curve and learning rate schedule is shown in Figure 6.

Table 6: Prompt template used for distillation (also referred to as the "Qwen2.5-math-cot" template). {question} represents each question.

| Prompt Template |
| --- |
| <lim_start\|>system
Please reason step by step, and put your final answer within \boxed{}. <lim_end\|>
<lim_start\|>user
{question} <lim_end\|>
<lim_start\|>assistant |

Table 7: Training configuration for distillation.

| Parameter | Value |
| --- | --- |
| Number of GPUs | $16 \times$ A800 |
| Total epochs | 5 |
| Total training step | 295 |
| Global batch size | 16 |
| Gradient accumulation steps | 1 |
| Block size (Max length) | 16,384 |
| Learning rate warmup ratio | 0.05 |
| Learning rate | 1e-5 |
| Learning rate scheduler | consine |
| Weight decay | 1e-4 |
| Adam beta1 / beta2 | 0.9 / 0.95 |

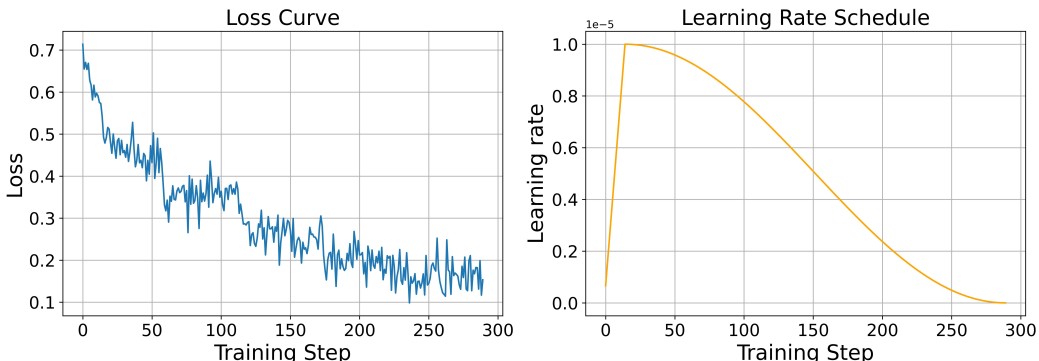

Figure 6: Training curves for our distillation.

### B.3 COMPARISON OF COMPUTATIONAL COSTS BETWEEN DISTILLATION AND ZERO-RL

Table 8: Comparison of computational costs between distillation and zero-RL.

| Computational Costs | Distillation 32B (Ours) | Zero-RL 32B |
| --- | --- | --- |
| Requirements of GPUs | $< 16 \times$ A800/H800 | Typically $> 64 \times$ A800/H800 |
| Training time | $< 3s$ hours | Typically $> 48$ hours |
| # of training samples | 920 | Typically $> 10,000$ |
| Performance | See Table 1 | See Table 1 |

As shown in Table 1 and Section 3, although the number of training samples for distillation and zero-RL is not directly comparable, there is a substantial difference in computational cost. As presented in Table 8, zero-RL typically requires several times, or even tens of times, more GPUs and training time than distillation. If we want to achieve better results with zero-RL, it would often require substantially more resources than those listed in Table 8. Note that as the number of distillation examples increases, the performance of the distilled model may continue to improve. For example, (Guo et al., 2025)

demonstrates that performing SFT with 800,000 examples can significantly enhance the base model's performance.

# C  DETAILS AND MORE RESULTS

As pointed out in (Hochlehnert et al., 2025), many detailed evaluation parameters can influence the results, especially on datasets like AIME or HMMT, which contain only 30 problems each. To ensure reproducibility, we report detailed evaluation settings in C.1 and include additional results under other parameters in C.2.

## C.1  EVALUATION DETAILS

**Evaluation setting.** In the main evaluation experiments (Section 3), all models are evaluated with a temperature of 1, a top-p of 0.95, a seed of 0, and a maximum generation length of 32,768. For open-source zero-RL models, we use the prompt templates specified in their original papers or reports (huggingface page). Specifically, the prompt template for DAPO-32B is shown in Table 10; the prompt template for ORZ-32B is shown in Table 9; the prompt template for SimpleRL-32B is shown in Table 11; the prompt template for Qwen2.5-32B-Base is shown in Table 12. For all benchmarks, we use the zero-shot setting.

**Evaluation framework.** As pointed out in (Hochlehnert et al., 2025), the choice of evaluation framework can even affect the results a lot. For fairness, all models are evaluated using the *same* evaluation framework. Specifically, we adopt the framework from Qwen2.5-Math[4], which itself is adapted from Math-Evaluation-Harness[5].

In practice, we find that answer extraction strategies can significantly affect evaluation results. For example, the prompt template of DAPO-32B requires the model to output the final answer after the token "Answer:", but does not require the answer to be enclosed in \boxed. As a result, the Qwen2.5-Math evaluation framework, which prioritizes extracting answers from within \boxed, may lead to inconsistencies in such cases. To accommodate these specific answer format requirements, we adapt the answer extraction strategy accordingly. For example, for DAPO-32B, we extract the text following "Answer:" as the final answer.

Table 9: Prompt template for DAPO-32B evaluation.

| Prompt Template |
| --- |
| <lim_startl>user
Solve the following math problem step by step. The last line of your response should be of the form Answer: $Answer (without quotes) where $Answer is the answer to the problem.

{question}

Remember to put your answer on its own line after "Answer:".<lim_endl>
<lim_startl>assistant |

## C.2  MORE RESULTS

**Distillation results using other teacher models.** As shown in Table 13, different teacher models yield different outcomes. If the teacher (such as GPT-4o) lacks flexible reasoning ability (advanced cognitive behaviors), distillation brings little benefit; but if the teacher (such as QwQ-32B and DeepSeek R1) frequently shows such patterns, the distilled model consistently improves. This provides additional evidence that these advanced cognitive behaviors play a critical role in reasoning.

**Evaluation results under lower temperature.** In Section 3 (Table 1), we set the temperature to 1 and evaluate the models using Avg@32 or Avg@8. Here, we additionally evaluate with a lower temperature of 0.6. The results are shown in Table 14. As shown, the evaluation results

---

[4]https://github.com/QwenLM/Qwen2.5-Math
[5]https://github.com/ZubinGou/math-evaluation-harness

Table 10: Prompt template for ORZ-32B evaluation.

| Prompt Template |
|---|
| <\|im_start\|>system
A conversation between User and Assistant. The User asks a question, and the Assistant solves it. The Assistant first thinks about the reasoning process in the mind and then provides the User with the answer. The reasoning process is enclosed within <think> </think> and answer is enclosed within <answer> </answer> tags, respectively, i.e., <think> reasoning process here </think> <answer> answer here </answer>.<\|im_end\|>.
<\|im_start\|>user
{question}<\|im_end\|>
<\|im_start\|>assistant
<think> |

Table 11: Prompt template for SimpleRL-32B evaluation (also referred to as the "Qwen-boxed" template.

| Prompt Template |
|---|
| <\|im_start\|>system
You are a helpful assistant.<\|im_end\|>
<\|im_start\|>user
{question}
Please reason step by step, and put your final answer within \boxed{}. <\|im_end\|>
<\|im_start\|>assistant |

Table 12: Prompt template for Qwen2.5-32B-Base evaluation. We use no template, as using no template leads to the best performance for Qwen2.5-32B-Base. See Table 15 for detailed comparison.

| Prompt Template |
|---|
| {question} |

Table 13: Performance of distilled models from different teacher models.

| Metric | Distilled-32B
(DeepSeek R1) | Distilled-32B
(GPT-4o) | Distilled-32B
(QwQ-32B) |
|---|---|---|---|
| AIME2024 (Avg@32) | 61.2 | 12.5 | 66.7 |
| AIME2025 (Avg@32) | 50.0 | 11.8 | 64.6 |
| HMMT Feb 2025 (Avg@32) | 34.6 | 3.75 | 42.5 |

under temperature 0.6 are similar to those under temperature 1, and our distilled model still clearly outperforms all other models across all benchmarks.

Table 14: Performance of different models across benchmarks at a lower temperature of 0.6.

| Metric | Distilled
-32B | Zero-RL
(DAPO-32B) | Zero-RL
(ORZ-32B) | Zero-RL
(SimpleRL-32B) | Qwen2.5
-32B-base |
|---|---|---|---|---|---|
| AIME2024 (Avg@32) | **59.3** | 51.3 | 44.4 | 28.6 | 20.1 |
| AIME2025 (Avg@32) | **49.2** | 34.8 | 34.5 | 9.4 | 9.8 |
| HMMT Feb 2025 (Avg@32) | **34.9** | 13.4 | 19.8 | 5.4 | 2.3 |
| GPQA Diamond (Avg@8) | **60.2** | 49.5 | 55.3 | 47.3 | 41.1 |
| MATH500 (Avg@8) | **94.1** | 67.2 | 90.8 | 82.4 | 75.2 |

**Different prompt templates affect the performance of Qwen2.5-32B-Base.** In Section 3 (Table 1), we report the performance of the base model Qwen2.5-32B-Base using no template. Interestingly, we find that different prompt templates can significantly affect the evaluation results of Qwen2.5-32B-Base, as shown in Table 15. Similar findings have also been reported for the Qwen2.5-Math base

model (Liu et al., 2025). The "no template" refers to the template in Table 12. The "Qwen-boxed template" refers to the template in Table 11. The "Qwen2.5-math-cot template" refers to the template in Table 6.

Table 15: Performance of Qwen2.5-32B-Base using different prompt templates. No template clearly outperforms other prompt templates. No template refers to the template in Table 12, Qwen-boxed template refers to the template in Table 11, Qwen2.5-math-cot template refers to the template in Table 6.

| Metric | Qwen2.5-32B-Base | | |
|---|---|---|---|
| | No template | Qwen-boxed template | Qwen2.5-math-cot template |
| AIME2024 (Avg@32) | **16.8** | 4.7 | 5.8 |
| AIME2025 (Avg@32) | **8.3** | 2.9 | 1.7 |
| HMMT Feb 2025 (Avg@32) | **1.9** | 0.5 | 0.5 |
| GPQA Diamond (Avg@8) | **34.9** | 34.9 | 32.3 |
| MATH500 (Avg@8) | **70.1** | 46.8 | 41.7 |

**Performance of the distilled and zero-RL models on other domains.** In addition to the complex mathematical reasoning tasks reported in Table 1, we also present the performance of the distilled and zero-RL models on other domains in Table 16. MMLU-Pro (Wang et al., 2024) consists of 12K complex questions spanning a wide range of disciplines such as Math, Physics, Chemistry, Law, Economics and Psychology. MMLU-STEM is a subset of the MMLU dataset (Hendrycks et al., 2020) focused specifically on STEM-related subjects. GPQA Diamond is also a science task, and we reuse the results from Table 1. For all benchmarks, we use the zero-shot setting.

As shown in Table 16, our distilled model also outperforms other models and performs strongly. Although the distilled data only contain mathematical content, the model's performance on general tasks does not show degradation and even benefit from the distillation beyond its original domain.

Table 16: Performance of different models across benchmarks in other domains.

| Metric | Distilled -32B | Zero-RL (DAPO-32B) | Zero-RL (ORZ-32B) | Zero-RL (SimpleRL-32B) | Qwen2.5 -32B-base |
|---|---|---|---|---|---|
| *General Tasks* | | | | | |
| MMLU-pro | **75.1** | 62.9 | 70.9 | 69.5 | 52.3[*] |
| *Science Tasks* | | | | | |
| GPQA Diamond | **60.0** | 48.7 | 57.7 | 48.4 | 34.9 |
| MMLU-stem | **91.0** | 88.6 | 89.6 | 85.7 | 76.9[*] |

[*] The slightly lower evaluation results compared to those reported in the Qwen2.5 technical report (Yang et al., 2024a) may be due to our use of *zero-shot* evaluation. For fairness, we report our evaluation results here where all models are evaluated using the same parameters.

For some general tasks, we observe similar potential reward hacking or overfitting phenomena in zero-RL output as discussed in Section 5. We select one example in Table 17. In this example, DAPO selects the correct answer, but ends up outputting an unrelated integer: 3, instead of the correct option.

## C.3   LLM FOR WRITING

The LLM was used only for light language polishing and grammar checking when writing this paper.

Table 17: An example of zero-RL outputs to a general task problem

**Question:** An astronomer observes that a planet rotates faster after a meteorite impact. Which is the most likely effect of this increase in rotation? (A) Planetary density will decrease. (B) Planetary years will become longer. (C) Planetary days will become shorter. (D) Planetary gravity will become stronger.
Please write your final answer in the form of \boxed{A}, \boxed{B}, \boxed{C}, \boxed{D}.

**Zero-RL model (DAPO-32B) response:**
To approach this math and physics problem step by step...

...
- Option (C): Planetary days will become shorter. Correct, as an increased rotation rate means the time for one rotation (i.e., a day) becomes shorter.
- Option (D): Planetary gravity will become stronger. Incorrect, as the effect is likely to make gravity weaker due to increased centrifugal force.
Therefore, the most likely effect of the planet rotating faster after a meteorite impact is that planetary days will become shorter.

Answer: $\boxed{C}$

Thus the correct answer is $\boxed{C}$

Answer: **3**

# D   MORE ANALYSIS ON LINGUISTIC PATTERNS AND ADVANCED COGNITIVE BEHAVIORS

## D.1   CONTRASTING SOLUTION STYLES OF DISTILLED AND ZERO-RL MODEL

We provide additional response comparisons between our distilled model and the zero-RL models in Tables 18, 19, and 20. Regardless of which zero-RL model is used, there are clear differences between the two types of models. Zero-RL models tend to follow a fixed and pre-defined reasoning path, while the distilled model frequently shifts its thinking and exhibits more awareness of potential mistakes.

It is worth noting that zero-RL models can indeed exhibit thinking shifts and awareness of mistakes, and sometimes even multiple times in some problems. However, compared to the distilled model, the overall frequency of such behaviors remains significantly *lower*. This observation aligns with the results in Figure 4, which show that the distilled model can exhibit the two advanced cognitive behaviors much more frequently, and both the zero-RL and base models are also capable of exhibiting them but to a *significantly lesser* extent.

Table 18: The contrasting solution styles of the two models on an example from AIME 2024.

**Question:** Let $\mathcal{B}$ be the set of rectangular boxes with surface area 54 and volume 23. Let $r$ be the radius of the smallest sphere that can contain each of the rectangular boxes that are elements of $\mathcal{B}$. The value of $r^2$ can be written as $\frac{p}{q}$, where $p$ and $q$ are relatively prime positive integers. Find $p + q$.

| **Zero-RL model (DAPO-32B) response:** | **Distilled model response:** |
|---|---|
| To approach this math problem step by step, we need to find... | Okay, so I need to find... |
| ... | ... |
| Therefore, we have two cases to consider from the equations $(b - a)(\lambda c + \mu) = 0$ and $(c - b)(\lambda a + \mu) = 0$. | **But wait**, actually, since all boxes in set B... |
| Case 1: $a = b = c$. | ... |
| ... | **Alternatively**, minimize... |
| Case 2: We have two variables equal while the third differs... | **But maybe there's a smarter way** to approach this. **Maybe** using Lagrange multipliers**?** |
| ... | ... |
| so, | This seems complicated, but **perhaps** we can find a relation between $l, w, h$... |
| $$\frac{3\sqrt{73}}{8} \approx \frac{3 \times 8.54}{8} = \frac{25.62}{8} \approx 3.20.$$ | ... |
| ... | **But wait, maybe** there are other configurations where the box is not even with two sides equal? |
| thus | **Alternatively, perhaps there is a mistake here**. **Wait**, if we... |
| $$\frac{\sqrt{17292.6}}{46} \approx \frac{131.5}{46} \approx 2.86.$$ | ... |
| Therefore, the radius for the configuration with... | But earlier, when I thought we needed to minimize the space diagonal, **I was wrong**... |
| ... | ... |
| [Answer]. | [Answer]. |

Table 19: The contrasting solution styles of the two models on an example from AIME 2024.

**Question:** Let $\triangle ABC$ have circumcenter $O$ and incenter $I$ with $\overline{IA} \perp \overline{OI}$, circumradius 13, and inradius 6. Find $AB \cdot AC$.

| Zero-RL model (ORZ-32B) response: | Distilled model response: |
|---|---|
| `<think>` | Okay, let's tackle this geometry problem... |
| Let's break down the problem step-by-step... | ... |
| ... | **I remember** that in triangle geometry, there are formulas relating the |
| **Step 1**: Interpret the condition $\overline{IA} \perp \overline{OI}$ | sides, circumradius, inradius, and distances between centers. **Maybe** |
| ... | Euler's formula relates some of these elements. |
| **Step 2**: Recall important properties and coordinates | ... |
| ... | **But wait**, let me verify that. Yes, **I think that's correct**... |
| **Step 3**: Consider Euler's formula relating the circumcenter and incen- | ... |
| ter | **Alternatively**, using trigonometric identities or properties related to |
| ... | the incenter and circumcenter. **Wait, another thought**: In triangle |
| **Step 36**: Going back to the coordinates and distance | geometry... |
| ... | ... |
| **Step 37**: Using the distances in terms of angles | **But perhaps that's not helpful** here... |
| ... | ... |
| Since $p$ and $q$ are positive (as they are products of magnitudes), the | **Wait, another approach**: since $IA$ is perpendicular to $OI$... |
| terms $(1 + \sqrt{1 - 4p^2})$ and $(1 + \sqrt{1 - 4q^2})$ are | ... |
| `</think> <answer>` | Therefore, the answer should be 468. Let me check if there's any |
| [Answer]. | possibility that **we made an error** in assuming $s = (3/2)a$. |
| `</answer>` | ... |
| | [Answer]. |

Table 20: The contrasting solution styles of the two models on an example from AIME 2025.

**Question:** The set of points in 3-dimensional coordinate space that lie in the plane $x + y + z = 75$ whose coordinates satisfy the inequalities $x - yz < y - zx < z - xy$ forms three disjoint convex regions. Exactly one of those regions has finite area. The area of this finite region can be expressed in the form $a\sqrt{b}$, where $a$ and $b$ are positive integers and $b$ is not divisible by the square of any prime. Find $a + b$.

| Zero-RL model (DAPO-32B) response: | Distilled model response: |
|---|---|
| Let's break down the problem step by step. | Okay, let's try to tackle this problem... |
| ... | ... |
| **Step 1**: Understanding the Problem | First, the boundaries of the inequalities are the lines where the prod- |
| ... | ucts are zero, i.e., the lines.. |
| **Step 2**: Analyzing the Inequalities | ... |
| ... | **Let me check** each combination. First, $A \cap C$: Region where |
| **Step 3**: Constraints on $x$, $y$, and $z$ | $x > y, x + y > 76, x < -1, x + 2y > 75$. **Is this possible?**... |
| We need to be mindful of the signs of $1 + z$ and $1 + x$ in the | Therefore, $x + y > 76$ is impossible here. Hence, $A \cap C$ is empty. |
| inequalities above. | ... |
| ... | Next, $A \cap D$: $x > y, x + y > 76, x > -1, x + 2y < 75$. |
| **Step 10**: Checking for Finite Area | **Possible?**... So, yes, this region exists. So, $A \cap D$ is non-empty. |
| #### For Case (Ib) and (IIa): | ... |
| ... | **Alternatively, maybe** the finite region is bounded in 3D space on the |
| #### For Case (Ib) and (IIb): | plane... |
| ... | ... |
| Let's **double-check** our triangle vertices and calculations to ensure... | **Wait, let's re-examine**. |
| ... | ... |
| `</think> <answer>` | **But perhaps this is similar to the previous approach**. **Alterna-** |
| [Answer]. | **tively**, consider normalizing the coordinates. |
| `</answer>` | ... |
| | [Answer]. |

## D.2 TOKEN CATEGORIZATION

By examining the outputs of our distilled and zero-RL models, we define three token categories based on their semantic meaning and role in the sentence. Different from (Yang et al., 2025), we modify the set of anthropomorphic tokens and redefine the categories of logical connectors and mathematical reasoning tokens. The full tokens categories are as follow:

- **Anthropomorphic tokens**: okay, me, hmm, aha, wait, hold on, yes, mistake, perhaps, maybe.

- **Logical connectors**: but, since, thus, however, because, therefore, so, alternatively, another.

- **Mathematical reasoning tokens**: assume, suppose, define, expand, apply, use, multiply, solve, simplify, substitute, combine, rewrite, equivalently, denote, rearrange, formula, plug, imply, follow, calculate, notice, expression, divide, add, start, set, evaluate, verify, check.

For each token, the frequency statistics include the token itself as well as its variants. For example, for the token "assume", we count both "assume" and "assuming", and report their combined frequency in Figure 1, 2 and 3.

For anthropomorphic tokens, we include words that are often found in real human conversations. Tokens such as "perhaps" and "maybe" are included because they express uncertainty, a trait frequently observed in human dialogue. We also include token "mistake" since it frequently shows up in sentences with conversational tone and usually indicates the speaker has recognized an error. For logical connectors, we select some common connectors that signal contrast, progression, or coordination in problem-solving process. For mathematical reasoning tokens, we include mathematical tokens that frequently appear in the outputs of both the zero-RL and distilled models.

### D.3 OUTPUT OF TOKEN-RESTRICTED DISTILLED MODEL

As shown in Subsection 4.1, certain tokens (specifically anthropomorphic tokens and some logical connectors) exhibit clear frequency differences between the distilled model and the zero-RL model. The distilled model produces significantly more of these distinctive tokens compared to the zero-RL model. As shown in subsections 4.1 and 4.2, preventing our distilled model from generating these distinctive tokens leads to a clear drop in both performance and the frequency of the two advanced cognitive behaviors. Specifically, we select the following banned tokens: "wait", "me", "perhaps", "maybe", "alternatively", "but", "another", "hold on", "hmm", "alternate", "alternately", "not sure", "okay", "seems", "though", "however". Apart from restricting the generation of these tokens, all other evaluation settings remain unchanged.

### D.4 EXPERIMENTS ABOUT ANALYZING ADVANCED COGNITIVE BEHAVIORS

**Experiments settings.** We prompt GPT-4o[6] to identify which parts of each response reflect either of the two advanced cognitive behaviors, and count how many times each behavior appears per response. The prompt template is shown in Table 21. Since LLM-as-a-judge evaluation may exhibit some instability, we mitigate this by sampling multiple times. For the AIME, GPQA and HMMT benchmarks, we randomly sample 4 responses per problem for each model and average the results. For the MATH500 benchmark, we sample 2 responses per problem for each model and average the results.

**More results.** We additionally include the statistics of two advanced cognitive behavior counts on the HMMT Feb 2025 benchmark in Figure 7.

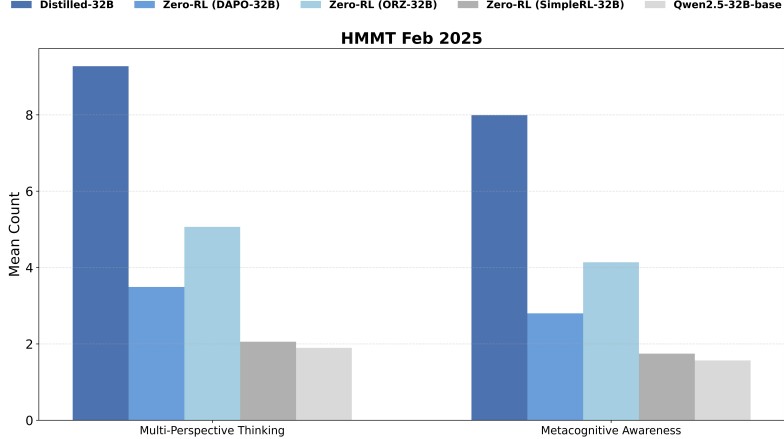

Figure 7: Comparison of the number of advanced cognitive behaviors per response on HMMT Feb 2025.

---

[6]The version used is GPT-4o-2024-05-13

Table 21: Prompt template for GPT-4o to identify the two advanced cognitive behaviors.

---

**Prompt Template**

---

In the process of solving difficult math problems, there are two types of advanced cognitive behaviors:

1. *Multi-Perspective Thinking or Attempting*: Viewing a problem from diverse perspectives to gain fresh insights, or exploring different ideas and alternative approaches to make meaningful progress. For example, expressions like "let's try another angle..." and "but I need a better strategy ... here's an idea, let's try...".

2. *Metacognitive Awareness*: Actively reflecting on your reasoning process during problem-solving to assess progress, evaluate current strategies, and identify potential errors in real time. Any reflective hesitation, backtracking, and verification are indicative of this awareness. For example, expressions like "wait, maybe my approach is wrong here" and "it seems not correct, step back".

Problem: {question}

Response: {response}

Based on the above response, please strictly identify whether the two advanced cognitive behaviors appear. Please think step by step, and finally output the relevant excerpts and the number of occurrences in a clean JSON format as shown below:

### JSON Output:

```
{
  "Multi-Perspective Thinking or Attempting": {
    "count": <number>,
    "excerpts": ["..."]
  },
  "Metacognitive Awareness": {
    "count": <number>,
    "excerpts": ["..."]
  }
}
```

---

# E   MORE DISCUSSION

**Why larger models can exhibit sustained performance improvements?** In this paper, we focus on *smaller* models (e.g., 32B) and highlight how distillation can enhance two advanced cognitive behaviors, enabling flexible reasoning and thereby improving overall reasoning performance. However, as shown in (Guo et al., 2025), the performing zero-RL on *larger* base model (DeepSeek-V3-Base 671B) can lead to substantial performance gains and the emergence of self-reflection reasoning capabilities. The outputs of DeepSeek-R1-Zero also contain the distinctive tokens emphasized in this paper (anthropomorphic tokens and some logical connectors), which contrasts with the rigid reasoning observed in zero-RL models trained on *smaller* models. For this issue, we propose two possible reasons. One possible reason is that, as pointed out by some studies (Liu et al., 2025), the larger base models already exhibit self-reflective keywords. This suggests that the two advanced cognitive behaviors discussed in this paper may already exist in the larger base model to a non-negligible extent. The second possible reason is that DeepSeek-R1-Zero may benefit from well-designed prompts, a robust training framework, and carefully tuned parameters during RL training. Combined with the stronger contextual understanding and reasoning ability of *larger* base models, this allows the model to recognize the importance of the two advanced cognitive behaviors before the output entropy becomes too low. The exact reasons behind this remain beyond the scope of this paper, and will need to be explored in future work.

**Constructing distillation data in the absence of a teacher model.** Beyond this work, we also try to construct distillation data in the absence of a teacher reasoning model. We select two responses from DeepSeek R1 (Guo et al., 2025) and Gemini2.5 Pro (Google DeepMind, 2025) as examples, and use two-shot prompting to guide GPT-4o to generate responses with similar patterns for different questions. We include the description of two advanced cognition behaviors in the prompt as well. However, possibly due to the excessive prompt length, GPT-4o still struggles to generate high-quality responses that exhibit the two advanced cognitive behaviors, and the resulting responses are shorter on

average compared to those from DeepSeek R1. Considering that only a small amount of distillation data is sufficient to activate these advanced cognitive behaviors, manually writing such examples may be a feasible alternative. We plan to explore this direction in future work.

