# OpenReview forum: "Why Distillation can Outperform Zero-RL: The Role of Flexible Reasoning"
_ICLR.cc/2026/Conference — ICLR 2026 Conference Withdrawn Submission_

### Official Review · Reviewer_mG5M · 2025-10-27

**Soundness:** 2
**Presentation:** 3
**Contribution:** 2
**Rating:** 4
**Confidence:** 4

**Summary:**

This work discuss the performance gap between zero-RL models and distilled models.  Authors adopt the popular open-sourcing zero-RL models compared to their own fine-tuned distilled model, and observe that distilled model significantly outperforms the zero-RL models. Authors consider that this performance gap is attributed to different token distributions; is that, the distilled model is more likely to think like human, containing more anthropomorphic tokens and logical tokens. Further analysis shows that even forbiddening these distinctive tokens, the distilled model can still perform comparable to zero-RL models. Authors also analyze the advanced cognitive behaviors, i.e. multi-perspective thinking or attempting and metacognitive awareness, and find that the distilled model consistently perform better.

**Strengths:**

This work studies an interesting topic of comparing distillation and zero-RL training.

Authors conduct in-depth analysis to explore why distillation is better to zero-RL training.

Good writing work.

**Weaknesses:**

1. Insufficient evaluations. This work only evaluates on 32B models. More evaluations should be conducted on smaller size models and other model series, like Qwen-2.5/3-7B/8B/14B or LLaMA models. All experiments conducted in this work are based on Qwen2.5-32B, which cannot reflect the generaliablity of the analysis.

2. I am a bit confused on the motivation of the work. It is good to reveal that the distilled model is better to zero-RL models for Qwen2.5-32B series models. But, does it commonly stand for most models, distillation methods and zero-RL methods? I do not catch any new techniques or insightful observations from the work.

3. How about the effects of the amount of training data and different teacher models?

**Questions:**

See weakness.

Line 411: Wwhen -> When

---

### Official Review · Reviewer_hGih · 2025-10-28

**Soundness:** 2
**Presentation:** 2
**Contribution:** 2
**Rating:** 2
**Confidence:** 4

**Summary:**

This paper examines whether small-scale reasoning distillation can outperform zero-shot Reinforcement Learning (RL) and explores the underlying reasons. The authors conducted a controlled experiment using the Qwen 2.5 32B model, comparing its distillation with a teacher against three distinct RL strategies. They then analyzed two significant linguistic patterns: anthropomorphic patterns and logical connectors and used them to explain why distillation fosters these while zero-shot RL does not. Finally, the paper discusses potential contributing factors such as reward hacking and overfitting.

**Strengths:**

- The framing of the paper is clear where the authors talk about a hypothesis (can distillation with limited samples outperform zero-RL) and then plan experiments in the direction to showcase their findings.
- The authors study the reasons behind it and found linguistic patterns and behaviours to justify the hypothesis.
- The experiments are controlled and the same model is used to test the hypothesis by training it with distillation vs zero-RL.
- The paper is readable and the conclusion is clear.

**Weaknesses:**

1. I think the paper central idea is known already to the community and that's why there is nothing new to get from the paper. It is well known that zero-RL is either quite hard to start (faces a cold start problem for small to mid sized models where the right answers is not presented in the rollouts) or quite expensive if started without SFT (Deepseek R1 paper mentions this briefly where they mentions zero-RL works but it would be better to warm it up with some samples, otherwise quite expensive at the start). The paper reaches to the exact same conclusion with Qwen 32B model (lets say a medium sized model) and there’s not a lot of takeaways.
2. If the authors wanted to go into the proposed hypothesis in detail, the experiments could have been designed better. A lot of different models with varied sizes could have been tested to show the scaling nature of the hypothesis, different sample sizes of distilled samples could have been used (100, 250, 500, …) and not just 920 directly and so on. This seems incomplete with the authors trying to make sense of the experiments by showing some linguistic phenomenon which is also not backed and seems like something would support the hypothesis no matter the final results.
3. GPT 4o as a teacher did not work while Deepseek R1 led to great performance. This means that the quality of data is very important and not necessarily means distillation is better than zero RL. Also the distillation data sample size per query is 10-16K tokens which involves backtracking, self reflection, improvement and so on. The same phenomenon in RL is more externally and I think unless the distillation is restricted or generated from a non-thinking model, this comparison is not fair and hence the conclusion of distillation > zero-RL is not clear.
4. Fix some typos like "consine", "scheduler", "3s hours", ..

**Questions:**

1. Can the authors provide a controlled experiment with limited tokens from distillation or samples generated from a non-thinking teacher and show that they reached to the same conclusion of distillation > zero-RL?
2. Can authors run experiments with different size of the dataset (100, 250, 500,..) and plot the curves to show when distillation gets better than RL?
3. Can authors run the banning tokens experiments with not just banning the tokens but rather rewriting it with some other LLMs and distilling with it? Banning a token reduces the prob of the entire sentence and hence breaks the flow and this way you can show style vs strategy difference? Are the tokens important or the style of that without those tokens?

---

### Official Review · Reviewer_Hxhi · 2025-10-30

**Soundness:** 3
**Presentation:** 3
**Contribution:** 2
**Rating:** 6
**Confidence:** 3

**Summary:**

This paper investigates whether distillation can outperform zero-RL for enhancing reasoning in smaller LLMs (<32B parameters). The authors fine-tune Qwen2.5-32B on only 920 AIME problems with responses generated by DeepSeek R1, and show this distilled model outperforms three state-of-the-art zero-RL models on mathematical reasoning benchmarks. Through linguistic analysis, they attribute this success to "flexible reasoning" characterized by increased use of anthropomorphic tokens (e.g., "wait," "maybe") and logical connectors (e.g., "alternatively," "but"). They further identify two "advanced cognitive behaviors": Multi-Perspective Thinking and Metacognitive Awareness, that appear more frequently in distilled model outputs and correlate with performance.

**Strengths:**

Empirical Contribution: The core finding that 920 distilled examples can match or exceed zero-RL models trained on 10-50× more data is practically valuable and challenges current assumptions about the necessity of expensive RL training for smaller models.

Teacher Model Ablation (Table 13): This is a strong experiment showing that distilling from GPT-4o (which lacks flexible reasoning patterns) provides minimal benefit while distilling from QwQ-32B and DeepSeek R1 works well. This supports the claim that the teacher's reasoning style, not just correctness, matters.

Comprehensive Evaluation: The paper includes multiple challenging benchmarks (AIME 2024/2025, HMMT, GPQA, MATH500) with careful attention to reproducibility details - fixed temperatures, multiple runs, detailed prompt templates, and unbiased Pass@k estimation.

Token Restriction Experiment: The ablation preventing generation of distinctive tokens (Table 3) is clever and provides evidence that these patterns matter for performance, even if the model attempts workarounds.

**Weaknesses:**

The core comparison is unfair and the paper's framing is a bit misleading. The title and abstract claim distillation "outperforms" zero-RL, but: Distillation uses DeepSeek R1, which itself required massive computational resources and RL training to develop. This is equivalent to comparing "learning from an expert's pre-computed solutions" versus "solving problems from scratch". The paper should compare total computational budgets including teacher training costs, not just student training. The authors acknowledge samples aren't "directly comparable" (lines 156-157) but still make superiority claims throughout. What you've actually shown is that transfer learning from an expensive teacher outperforms training from scratch with limited compute.

The "advanced cognitive behaviors" are standard problem-solving strategies people apply in complex problem solving. The contribution is showing distillation transfers advanced cognitive behaviors while zero-RL doesn't, which is descriptive rather than explanatory. The paper doesn't reveal why zero-RL fails to discover these patterns or how to induce them without distillation.

LLM-as-Judge Reliability: The cognitive behavior analysis (Section 4.2, Figure 4) relies entirely on GPT-4o judgments. The author should justify with human validation and robustness checks to show its reliability.

Narrow experimental scope: Single base model (Qwen2.5-32B) and single model family. Training on AIME (1983-2023), testing on AIME (2024-2025) creates distribution matching advantage. No cross-model family validation (Llama, Mistral, etc.)

**Questions:**

1. Can you provide a complete accounting of computational costs including DeepSeek R1's training? How does "920 examples of distillation + R1's training cost" compare to "zero-RL from scratch"?

2. Have you tried distillation followed by RL? Does this combination outperform either alone? This would directly test your hypothesis that distillation provides a better starting point for RL.

3. Can you have human annotators validate the GPT-4o cognitive behavior counts on a subset (e.g., 50 problems)? What is the inter-rater agreement?

---

### Official Review · Reviewer_5K35 · 2025-11-01

**Soundness:** 2
**Presentation:** 4
**Contribution:** 3
**Rating:** 6
**Confidence:** 4

**Summary:**

This paper investigates potential reasons on why distillation on a small amount of examples can outperform zero-RL for improving reasoning abilities of smaller language models (32B). The authors demonstrate that distilling from DeepSeek R1 onto Qwen2.5-32B-base substantially outperforms state-of-the-art zero-RL models across multiple challenging benchmarks. They find that distilled models exhibit more flexible reasoning patterns, characterized by higher frequency of anthropomorphic tokens and logical connectors.

**Strengths:**

- The paper addresses a practically relevant question about distillation versus zero-RL for smaller models, showing that distillation can outperform zero-RL in some scenarios. The findings have direct implications for practitioners.
- Token frequency analysis quantifies stylistic differences between approaches. The conceptualization of "Multi-Perspective Thinking" and "Metacognitive Awareness" offers a framework for understanding machine reasoning, and the token-restriction experiment demonstrates causality.
- The paper provides sufficient detail including training configurations, prompt templates, and appendices for reproducibility.

**Weaknesses:**

- The success of distillation depends on access to a superior teacher model (DeepSeek R1) with existing reasoning capability. Table 13 shows distilling from GPT-4o yields poor results. This prerequisite limits the method's scope to scenarios where such expert teachers exist, yet receives insufficient emphasis in the main text.
-  Several confounding factors complicate interpretation. The 920 AIME problems represent extremely difficult competition mathematics from a single domain—is the gain from flexible reasoning style, problem difficulty, or both? The distilled model also generates significantly longer responses (Table 1). How much improvement stems from extended thinking (longer CoT) versus the specific cognitive structure? Experiments with simpler problems or controlled response lengths could disentangle these effects.

**Questions:**

- How does performance change as the number of distillation examples varies (e.g., 100, 500, 920)?
- Figures 1-3: The font size is too small. The authors might want to increase font size for token labels in later versions.

---

### Note · Authors · 2026-01-06

**Comment:**

We thank all the reviewers for their thoughtful reviews.

**Withdrawal Confirmation:**

I have read and agree with the venue's withdrawal policy on behalf of myself and my co-authors.